# Research on Improved Wavelet Threshold Denoising Method for Non-Contact Force and Magnetic Signals

Xiaoxiao Li [1,*], Kexi Liao [2], Guoxi He [2] and Jianhua Zhao [2]

1   Sinopec Petroleum Engineering Zhongyuan Co., Ltd., Zhengzhou 450007, China
2   State Key Laboratory of Oil and Gas Reservoir Geology and Exploitation, Southwest Petroleum University, Chengdu 610500, China
*   Correspondence: 18382244018@163.com; Tel.: +86-183-8224-4018

**Abstract:** In order to solve the problem of noise interference in the collected magneto mechanical signals, a new wavelet shrinkage threshold based on adaptive estimation is proposed. Based on the shortcomings of the traditional threshold function, an improved threshold function is proposed, and the parameters of the threshold function are solved by the improved genetic algorithm, and the optimal denoising effect is finally obtained. The new threshold function can not only make up the defects of each threshold function, ensure the continuity of threshold function, but also flexibly adjust the threshold to adapt to different noise conditions, and solve the deviation caused by inherent threshold function, and protect the useful information with noise signals.

**Keywords:** force magnetic signal; wavelet denoising method; adaptive threshold; genetic algorithm; wavelet basis function

## 1. Introduction

The force and magnetic signals of metal components contain a lot of characteristic information. In order to extract useful feature information effectively, it is necessary to denoise the magnetic signal [1–3]. Fourier transform can not approximate the local information of motor vibration signal and is not suitable for noise reduction of magnetic signal [4,5]. The wavelet function has the function of local analysis, which can approach the detail characteristics of the signal very well, and is convenient for extracting the feature information of metal components [6]. The fast wavelet decomposition algorithm proposed in the literature makes the wavelet widely used in many fields. On this basis, wavelet threshold denoising has been developed rapidly [2,7–10].

The principle of wavelet threshold denoising algorithm is that signal and noise present different forms in wavelet domain [11,12]. With the increase of decomposition scale, the amplitude of noise figure decreases rapidly to zero, while the amplitude of real signal coefficient remains unchanged [4,8,13–15]. In the process of threshold denoising, the selection of threshold and threshold function is an important step [16]. At present, adaptive threshold is one of the most popular threshold selection methods [17–22].

According to the latest literature [17–21], the current research direction of wavelet threshold denoising mainly lies in the construction of a new threshold function and the selection of threshold. However, there are few research studies on wavelet basis function and decomposition level. There is no basis for the selection of wavelet basis function and the number of decomposition layers, so it is necessary to analyze and optimize the wavelet basis function and the number of decomposition layers. After selecting the best wavelet basis function and the number of decomposition layers, it is necessary to select the threshold function and threshold value.

In this paper, signal-to-noise ratio (SNR) and minimum mean square error (RMSE) are used as evaluation criteria. The optimization of wavelet basis function and decomposition level is carried out. SNR and RMSE were calculated for different wavelet basis

functions and decomposition levels. We select the wavelet basis function and the number of decomposition layers with the largest SNR value and the smallest RMSE.

For the selection criteria of threshold, this paper adopts the adaptive threshold algorithm to determine the threshold.

At the same time, the hard threshold function is a discontinuous function, and the reconstructed signal has oscillation. In addition, the soft threshold function has deviation from the real value of signal when the signal is contracted. In view of the above shortcomings, this paper proposes an improved threshold function algorithm, which uses the improved threshold function combined with the new threshold to modify the wavelet coefficients, and realizes the effective extraction of the signal.

## 2. Mathematical Background

It is assumed that $X = [x_0, x_1, x_2, \cdots, x_{N-1}]^T$ is an observation value containing noise. In other words [23–25],

$$x_i = s_i + n_i, (i = 1, 2, \cdots, N) \tag{1}$$

$s_i$ is the true value of the signal at time $i$, and $n_i$ is the independently distributed white Gaussian noise $N(0, \sigma)$. Our goal is to find an estimate $\hat{S}$ of the signal $S$ based on the observed value of $X$. To minimize the minimum mean squared error of $S$ and $\hat{S}$, we replace the mathematical expectation by the mean:

$$\xi(\hat{S}, S) = \frac{1}{N} \|\hat{S} - S\|^2 = \frac{1}{N} \sum_{i=1}^{N} (\hat{s}_i - s_i)^2 \tag{2}$$

The noise elimination method proposed by Donoho [17,26–29] is effective in the sense of minimum mean square error and can achieve better visual effects. Its main theoretical basis is that the signal belonging to Besov space is in the wavelet domain, and its energy is only concentrated in a limited number of coefficients. However, the noise energy is distributed in the whole wavelet domain. Therefore, the coefficient of signal after wavelet decomposition is larger than the coefficient of noise. Furthermore, the signal coefficients can be preserved, and most noise coefficients can be reduced to zero by using the threshold method.

From the above theory, as showm in Figure 1, we can reach the following wavelet thresholding denoising steps:

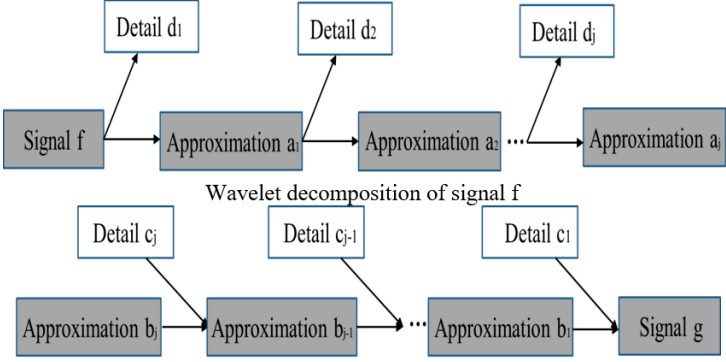

Figure 1. Signal reconstruction and decomposition process.

(1) The appropriate wavelet basis function is used to decompose the input noise signal, and the corresponding approximation coefficient and detail coefficient are extracted.

(2) Threshold wavelet coefficients. The wavelet coefficients of the original signal are retained, and other wavelet coefficients are eliminated.

(3) The threshold coefficient is reconstructed by iswt (iswt is the name of a function defined in MATLAB). In this way, the asymptotic estimation of the original signal can be obtained.

## 3. Adaptive Wavelet Threshold Denoising Method

### 3.1. Selection of Wavelet Basis Function

The selection principle of wavelet basis function is generally considered from the specific signal characteristics and wavelet basis function attributes. As shown in Table 1 Wavelet basis has the mathematical characteristics of orthogonality, symmetry, regularity, vanishing moment and tight support. The orthogonality reflects the degree of perfection of wavelet bases, and the wavelet bases with good orthogonality are conducive to the accurate reconstruction of wavelet decomposition coefficients. Symmetrical wavelet bases have linear phase, which can avoid phase distortion in signal decomposition and reconstruction. The higher the regular basis of wavelets is, the higher the regularity of the basis is. In order to detect the singular points effectively, the selected wavelet base must have enough vanishing moments. The larger the support, the better the regularity. The smaller the width of the support, the faster the calculation speed of wavelet transform.

**Table 1.** Parameter characteristics of wavelet basis functions.

| Wavelet Basis | Orthogonality | Biorthogonality | Symmetry | Compactness | Vanishing Moment | Support Length | Filter Length |
|---|---|---|---|---|---|---|---|
| Haar | Yes | Yes | Symmetric | Yes | 1 | 1 | 2 |
| Daubechies | Yes | Yes | Symmetric approximation | Yes | N | 2N-1 | 2N |
| Biorthogonal | No | Yes | Asymmetry | Yes | N-1 | Restructure 2Nr + 1 Decompose 2Nd + 1 | Restructure 2Nr + 2 Decompose 2Nd + 2 |
| Coiflet | Yes | Yes | Symmetric approximation | Yes | 2N | 6N-1 | 6N |
| Symlet | Yes | Yes | Symmetric approximation | Yes | 2N | 2N-1 | 2N |

As shown in Figure 2 and Table 2, we measured the surface of the pipe by instrumentMagnetic induction data are collected by three-dimensional magnetic signal sensors. The main sources of signal noise are electromagnetic components, transistors, resistors and integrated circuits. In addition, the buried depth and direction of buried pipeline, acquisition environment, geomagnetic field, and operator's own factors are also included. Secondly, as shown in Figure 3, there are some mutation points with large amplitude in the data, which are caused by many factors. These points can not be easily listed as noise signals. It is necessary to conduct many tests to determine the causes before making a decision. Therefore, we should not only remove the noise in the data as much as possible, but also retain some large value mutation points in the data.

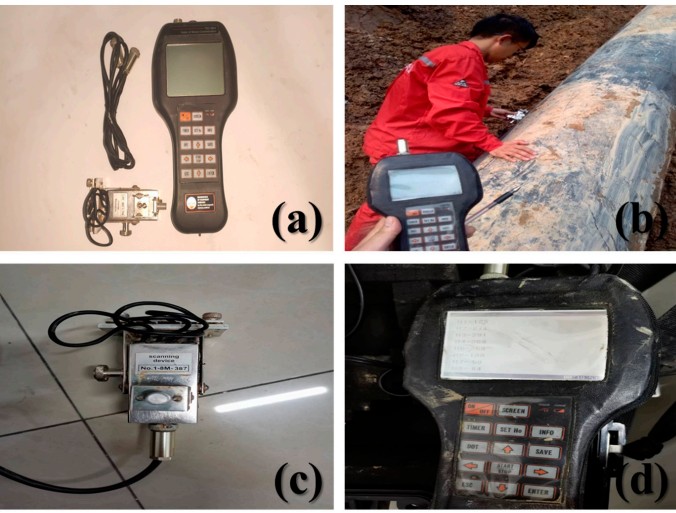

**Figure 2.** Non-contact 3D magnetic testing system. (**a**) The whole testing instrument; (**b**) Field operation; (**c**) data collector; (**d**) data analyzer.

**Table 2.** Technical parameters of tsc (Triaxial Stress Concentration)-2 m-8 tester.

| Parameter Name | Parameter |
|---|---|
| Hp range | ±2000 A/m |
| Hp measurement channel | 2–8 channel |
| Minimum measurement step width | 1 mm |
| Maximum measurement step width | 128 mm |
| Scanning speed | 0.2–0.5 m/s |
| Basic relative error | <5% |
| Relative error of measuring length | <5% |
| Microprocessor | 16 bit |
| Memory capacity | 1 Mb |
| Flash memory capacity | 32 Mb |
| Display | LCD, 320 × 240 |
| Transmission speed | 115 kbps |
| Keyboard | 14 keys |
| Battery | 7.2 V |
| Power consumption | 0.8–3.0 VA |
| Operating temperature range | −15–55 °C |
| Relative humidity range | 45–85% |
| Geometric dimension | 243 × 120 × 40 mm |
| Weight | 0.6 kg |

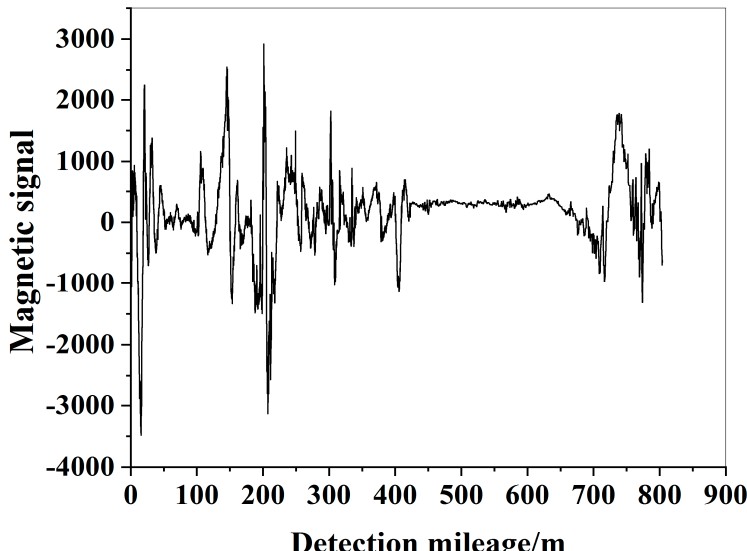

**Figure 3.** Magnetic signal data of sensor in a 1 z direction.

Through the above analysis, the wavelet basis function suitable for magnetic signal denoising should have good orthogonality, high vanishing moment, high regularity and moderate tight support length. Therefore, the approximate symmetric wavelet basis can be used. According to the characteristics of wavelet basis function parameters in the table, Daubechies (dbN), coifflet (coifN) and Symlet (symN) wavelet basis functions are suitable for magnetic signal data de-noising.

According to Figures 4 and 5, under different threshold functions, the Decomposition level is 5 to obtain SNR and RMSE diagrams under different wavelet basis functions. We find that DB7 has good denoising performance. Therefore, we select DB7 as wavelet basis function, and the schematic diagram of wavelet basis function is shown in Figures 6 and 7. We set the number of decomposition levels to 6, the threshold value to Rigrsure, and the threshold function to a soft threshold function.

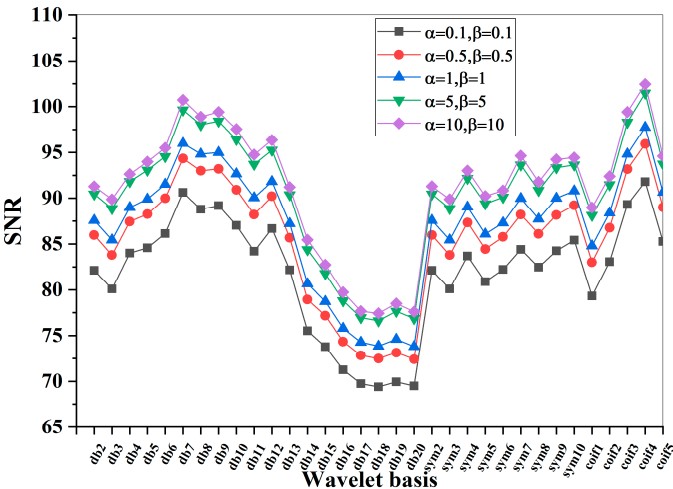

**Figure 4.** SNR values under different wavelet basis functions.

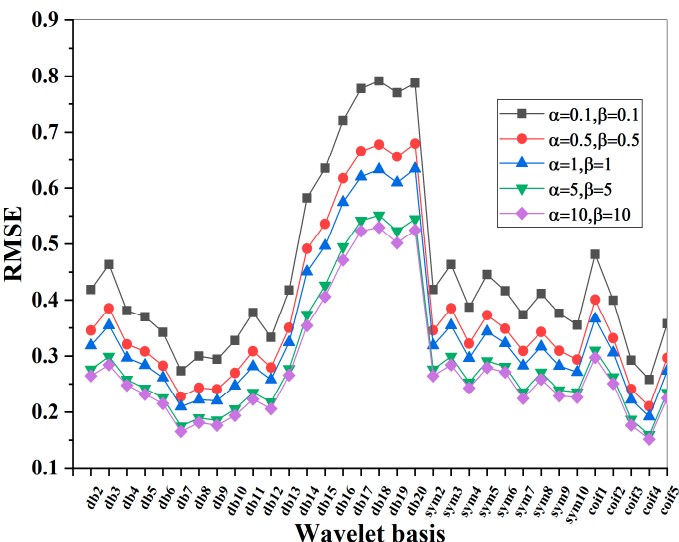

**Figure 5.** RMSE under different wavelet basis functions.

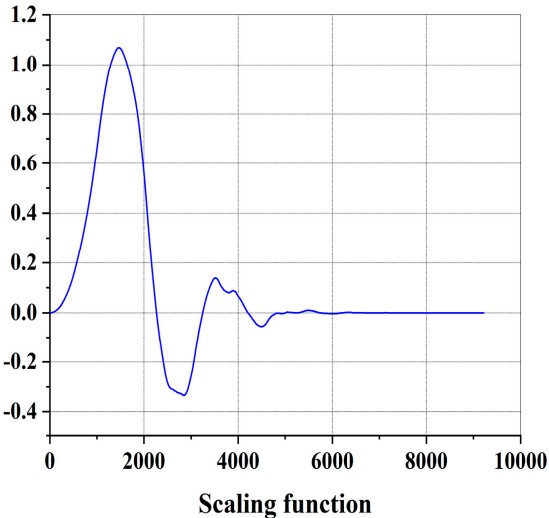

**Figure 6.** Scaling function.

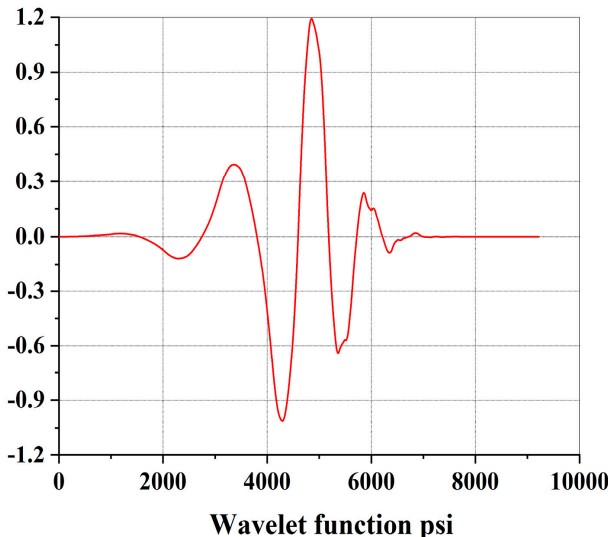

**Figure 7.** Wavelet function.

### 3.2. Determination of Decomposition Levels

For the selection of discrete wavelet decomposition layer, there is no systematic study and no strict basis for partitioning. Therefore, researchers need to try according to the actual problems. Different levels of decomposition will be used to analyze the signal after noise reduction, so as to determine the most suitable level of wavelet decomposition for this test. We select DB7 as the basis function, the threshold function as the hard threshold function, and the threshold selection method as Sqtwolog to verify the denoising effect.

The number of decomposition layers can not be determined by SNR, and the actual de-noising effect should be observed. On the one hand, the filtered noise is required to be clean, and on the other hand, the denoising effect should avoid the signal being over-filtered. Through comparison, it is found that, as shown in Figures 8–10, when the decomposition level is 5, the curve with decomposition level of 5 not only retains the overall trend information of the original signal, but also has a good approximation for local detail information, so the denoising quality is good. Therefore, we choose the five levels. The detail coefficients and approximate coefficients under different decomposition layers are obtained, as shown in Figures 11 and 12.

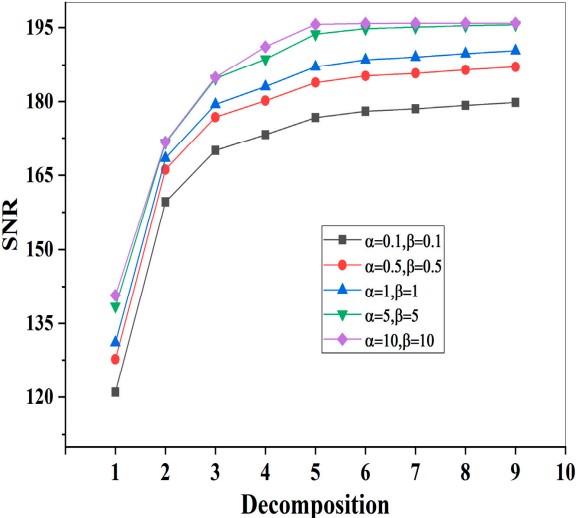

**Figure 8.** SNR under different decomposition levels.

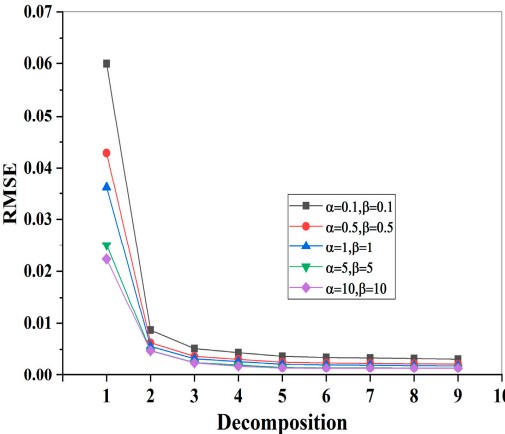

**Figure 9.** RMSE with different decomposition levels.

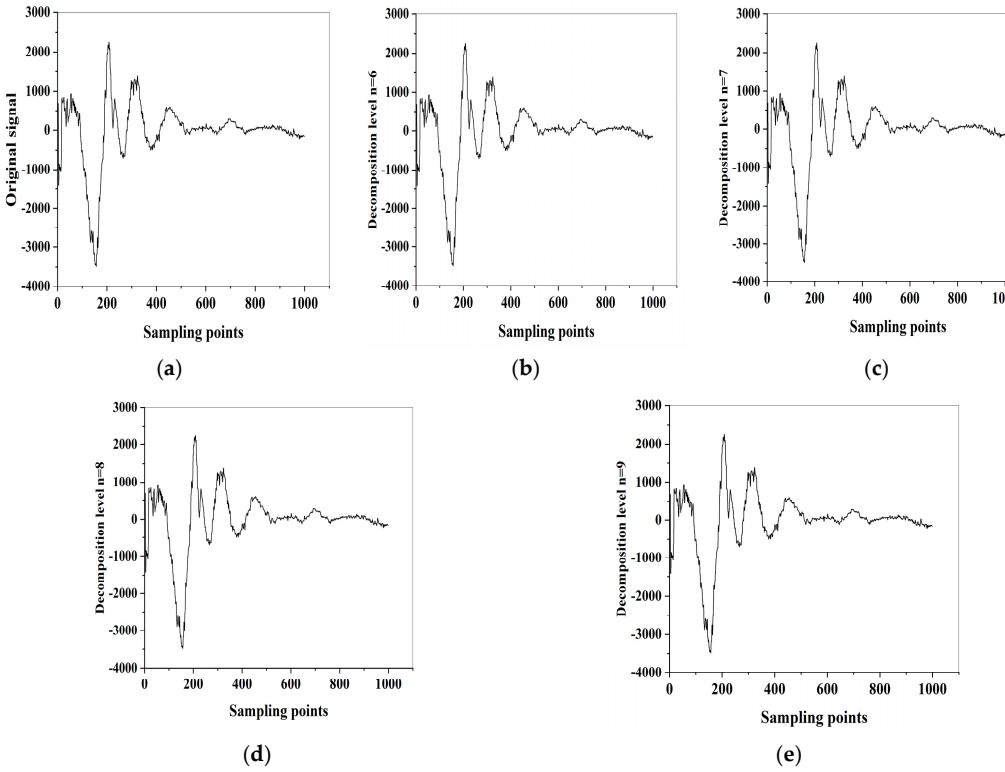

**Figure 10.** Denoised signals with different decomposition levels. (**a**) 5 levels; (**b**) 6 levels; (**c**) 7 levels; (**d**) 8 levels; (**e**) 9 levels.

### 3.3. Construction of Threshold Function

Hard threshold function expression [23–26]

$$
y_{k,j} = \begin{cases} \omega_{k,j}, & \left|\omega_{k,j}\right| \geq T \\ 0, & \left|\omega_{k,j}\right| < T \end{cases} \tag{3}
$$

Soft threshold function expression

$$
y_{k,j} = \begin{cases} \operatorname{sgn}(\omega_{k,j})\left(\left|\omega_{k,j}\right| - T\right), & \left|\omega_{k,j}\right| \geq T \\ 0, & \left|\omega_{k,j}\right| < T \end{cases} \tag{4}
$$

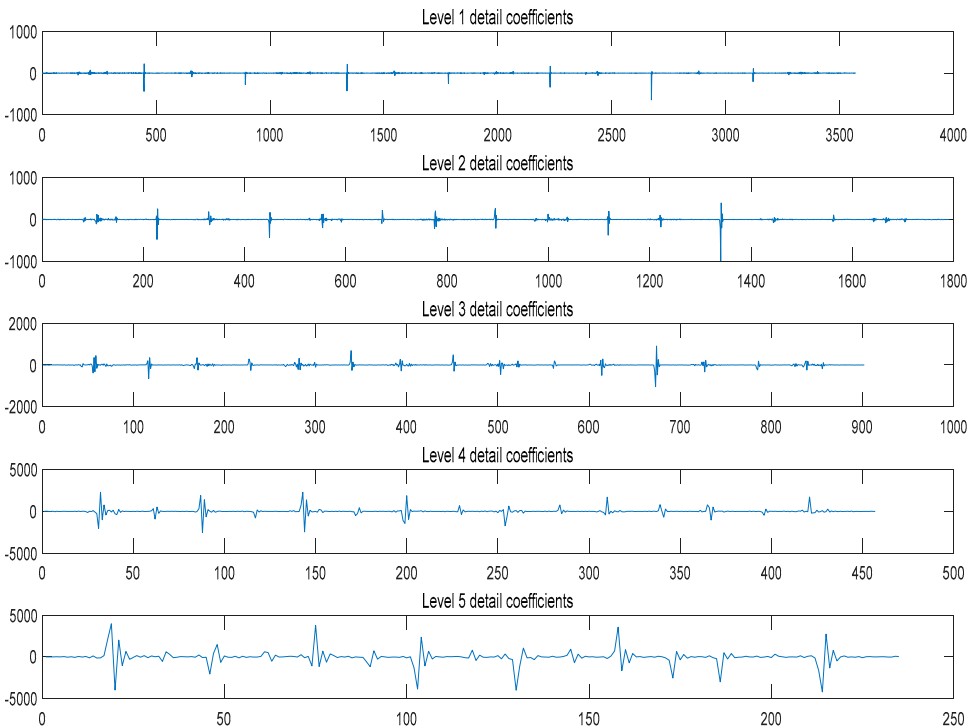

**Figure 11.** Detailed coefficient under different decomposition layers.

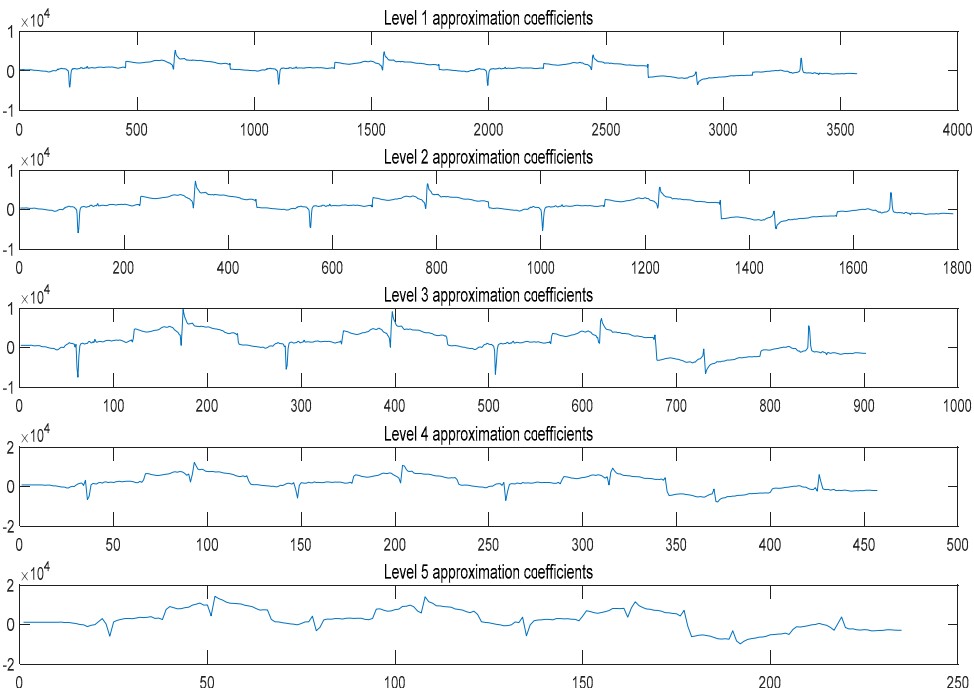

**Figure 12.** Approximate coefficients under different decomposition levels.

In the past wavelet threshold denoising, hard threshold function and soft threshold function are often used, but there are some problems such as fixed deviation and discontinuity. In order to solve this problem, the paper tries to use shape adjustment parameters to improve the threshold function, which can solve the disadvantages of soft and hard thresholds to a certain extent, and achieve a good denoising effect. Based on this, we

reconstruct a new threshold function according to the shortcomings of the two improved threshold functions in the literature:

$$y_{k,j} = \begin{cases} u\omega_{k,j} + (1-u)\mathrm{sgn}(\omega_{k,j})(\omega_{k,j} - \dfrac{2T}{1+e^{\frac{\beta(\omega_{k,j}-T)}{T}}}), \omega_{k,j} \geq T \\ 0, \left|\omega_{k,j}\right| < T \\ u\omega_{k,j} + (1-u)\mathrm{sgn}(\omega_{k,j})(-\omega_{k,j} - \dfrac{2T}{1+e^{\frac{\beta(-\omega_{k,j}-T)}{T}}}), \omega_{k,j} \leq -T \end{cases} \tag{5}$$

$$u = 1 - e^{-\alpha(\frac{|\omega_{k,j}|-T}{T})^2} \tag{6}$$

$$f = ux + (1-u)\mathrm{sgn}(x)(|x| - \frac{2T}{1 + e^{\frac{\beta(|x|-T)}{T}}}), \ \alpha > 0 \tag{7}$$

$$y = \frac{-2}{1 + e^{-\beta x}}, \beta \in R^+ \tag{8}$$

The constructed function curve is shown in Figure 13. The function value is distributed between $[-2, 0]$, and decreases with the increase of the $x$ value, and the slope of the curve increases from $-60$ to $60$.

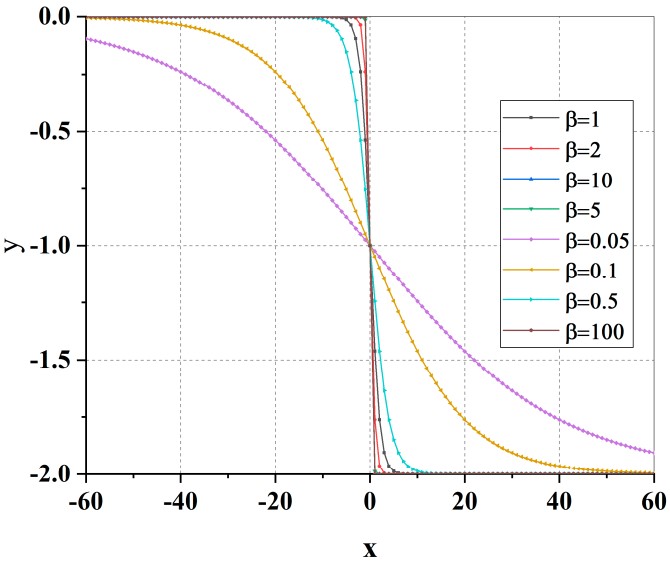

**Figure 13.** Constructed function curve.

*3.4. Feasibility Analysis of Constructed Function*

1. Function continuity

$$\lim_{\omega_{k,j}\to T^+} y_{k,j} = \lim_{\omega_{k,j}\to T^+} u\omega_{k,j} + (1-u)\mathrm{sgn}(\omega_{k,j})(\omega_{k,j} - \frac{2T}{1+e^{\frac{\beta(\omega_{k,j}-T)}{T}}}) = \lim_{\omega_{k,j}\to T^+}(1 - e^{-\alpha(\frac{\omega_{k,j}-T}{T})^2})T = 0 \tag{9}$$

In the same way:

$$\lim_{\omega_{k,j}\to T^-} y_{k,j} = 0 \tag{10}$$

Therefore, the threshold function is continuous at $T$.
Similarly, it can be proved that:

$$\lim_{\omega_{k,j}\to -T^-} y_{k,j} = \lim_{\omega_{k,j}\to -T^+} y_{k,j} = 0 \tag{11}$$

The threshold function is continuous at $-T$, and the function satisfies high-order differentiability.

2. The asymptotic property of function

$$\lim_{\omega_{k,j}\to-\infty}\frac{y_{k,j}}{\omega_{k,j}}=\lim_{\omega_{k,j}\to-\infty}\frac{u\omega_{k,j}-(1-u)(-\omega_{k,j}-\frac{2T}{1+e^{\frac{\beta(\omega_{k,j}-T)}{T}}})}{\omega_{k,j}}=\lim_{\omega_{k,j}\to-\infty}[2(1-e^{-\alpha(\frac{-\omega_{k,j}-T}{T})^{2}})-1]=1 \qquad (12)$$

In the same way:

$$\lim_{\omega_{k,j}\to+\infty}\frac{y_{k,j}}{\omega_{k,j}}=1 \qquad (13)$$

Asymptotic line of function:

$$y_{k,j}=\omega_{k,j} \qquad (14)$$

Therefore, when the wavelet coefficients are large, the fixed deviation problem of soft threshold function can be effectively reduced.

3. Deviation

$$\lim_{\omega_{k,j}\to+\infty}(y_{k,j}-\omega_{k,j})=\lim_{\omega_{k,j}\to+\infty}(u-1)\omega_{k,j}+(1-u)(\omega_{k,j}-\frac{2T}{1+e^{\frac{\beta(\omega_{k,j}-T)}{T}}})=0 \qquad (15)$$

In the same way, we can obtain that

$$\lim_{\omega_{k,j}\to-\infty}(y_{k,j}-\omega_{k,j})=0 \qquad (16)$$

4. Analysis of influence parameters of threshold function

When $\alpha=0$, $\beta=0$, the improved threshold function is a soft function; when $\alpha\to\infty$ or $\beta\to\infty$, the improved threshold function is a hard function. Therefore, the improved threshold function has good continuity at the threshold point. According to the parameters $\alpha$ and $\beta$, it can adapt to different signals.

5. Parity

The definition domain of the function is R and satisfies the requirement:

$$y_{k,j}(-\omega_{k,j})=-y_{k,j}(\omega_{k,j}) \qquad (17)$$

Therefore, the function is an odd function, which is consistent with the soft threshold and hard threshold.

As shown in Figure 14, the improved threshold function curve is distributed between the soft and hard threshold curves, which not only solves the problem of discontinuity at the threshold, but also relieves the energy loss caused by constant deviation.

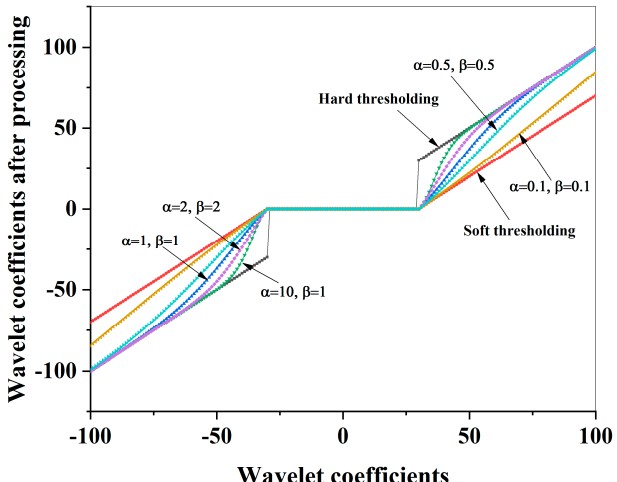

**Figure 14.** Comparison of hard threshold, soft threshold, and improved threshold function.

Through calculation, we obtain the SNR and RMSE values under different parameters, as shown in Figure 15. It can be seen from the figure that, when $\beta$ is known, with the increase of $\alpha$, SNR gradually increases and finally tends to a fixed value. On the contrary, RMSE gradually decreases and finally tends to a constant value. However, when $\alpha$ is known, the SNR value gradually increases with the increase of $\beta$, and finally tends to a constant value, while the change rule of RMSE is the opposite. Therefore, by adjusting the two parameters, we can obtain the ideal threshold function value, and then obtain the best wavelet coefficients and denoising effect.

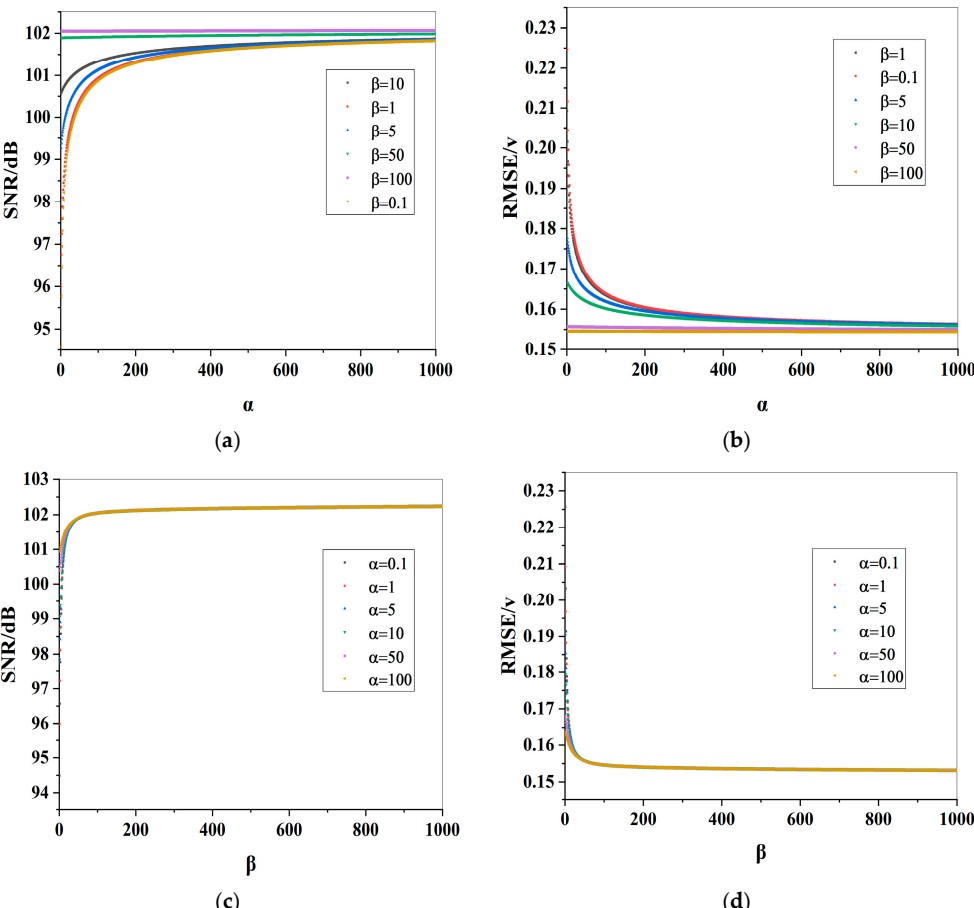

**Figure 15.** SNR and RMSE curves under different parameter values (**a**) SNR values under different $\alpha$; (**b**) RMSE values under different $\alpha$; (**c**) SNR values under different $\beta$; (**d**) RMSE values under different $\beta$.

### 3.5. Estimation of Threshold

The adaptive threshold algorithm is based on the steepest descent method in the optimization method. In other words, the threshold at the next time should be equal to the threshold at the present time plus a gradient value proportional to the negative mean square error function.

The algorithm is based on the steepest descent method in the optimization method, that is, the threshold $T(n+1)$ at the next time should be equal to the threshold $T(n)$ at the present time plus a gradient value $\Delta T(n)$ proportional to the negative mean square error function:

$$T(n+1) = T(n) - \mu \cdot \Delta T(n) \tag{18}$$

where $\mu$ is learning rate, and $\Delta T(n)$, and the expression is as follows:

$$\Delta T(n) = \frac{\partial \xi(n)}{\partial T(n)} \tag{19}$$

Therefore, the key to derive this algorithm is to find $\Delta T(n)$. We can set a function $g(\Omega)$ of the observed value, whose expression is:

$$g(\Omega) = \hat{s}(\Omega) - \Omega \tag{20}$$

If the valuation of signal $S$ is based on the observed value $\omega$, and thus $g(\Omega)$ belongs to the mapping from $R^N$ to $R^N$, and $g(\Omega)$ is differentiable, then:

$$\xi(\hat{S}, S) = E\left[\|\hat{s}(\Omega) - s\|^2\right] = N + E\left\{\|g(\Omega)\|^2 + 2\nabla_y \cdot g(\Omega)\right\} \tag{21}$$

$$\nabla_y \cdot g(\Omega) = \sum_1^N \frac{\partial g_i}{\partial \omega_{k,j}} \tag{22}$$

SURE is an unbiased estimate of the above mean square error, and its expression is as follows:

$$\xi(\hat{S}, S) = N + E\|g(\Omega)\|^2 + 2\nabla_y \cdot g(\Omega) \tag{23}$$

The gradient expression of the mean square error $\xi$ is:

$$\Delta T(n) = \frac{\partial \xi(n)}{\partial T(n)} = 2\sum_1^N g_i \cdot \frac{\partial g_i}{\partial T(n)} + 2\sum_1^N \frac{\partial^2 g_i}{\partial \omega_{k,j}\partial T(n)} \tag{24}$$

According to Equation (28),

$$g_i = y_{k,j} - \omega_{k,j} \tag{25}$$

A new threshold function (Equation (13)) is presented in this paper. It has continuous derivatives of infinite order. Based on Equation (32), an adaptive iterative operation is carried out to dynamically seek the best threshold. According to Equation (32), we can obtain:

$$\frac{\partial g(i)}{\partial T(n)} = \omega_{k,j}\frac{\partial u}{\partial T(n)} + \mathrm{sgn}(\omega_{k,j})(-\frac{\partial u}{\partial T(n)})(\omega_{k,j} - \frac{2T(n)}{1 + e^{\frac{\beta(\omega_{k,j}-T(n))}{T(n)}}}) + \mathrm{sgn}(\omega_{k,j})(1-u)\frac{-2(1 + e^{\frac{\beta(\omega_{k,j}-T(n))}{T(n)}}) + 2T(n)e^{\frac{\beta(\omega_{k,j}-T(n))}{T(n)}}(-\frac{\beta\omega_{k,j}}{T(n)^2})}{[1 + e^{\frac{\beta(\omega_{k,j}-T(n))}{T(n)}}]^2} \tag{26}$$

$$\frac{\partial u}{\partial T(n)} = 2\alpha\left(\frac{\omega_{k,j} - T(n)}{T(n)}\right)\left(\frac{\omega_{k,j}}{T(n)^2}\right)e^{-\alpha(\frac{\omega_{k,j}-T(n)}{T(n)})^2} \tag{27}$$

$$\frac{\partial g(i)}{\partial \omega_{k,j}} = u - 1 + \omega_{k,j}\frac{\partial u}{\partial \omega_{k,j}} + \left[(-\frac{\partial u}{\partial \omega_{k,j}})\mathrm{sgn}(\omega_{k,j})(\omega_{k,j} - \frac{2T(n)}{1 + e^{\frac{\beta(\omega_{k,j}-T(n))}{T(n)}}})\right] + (1-u)\mathrm{sgn}(\omega_{k,j})\left[1 - \frac{-2\beta e^{\frac{\beta(\omega_{k,j}-T(n))}{T(n)}}}{\left[1 + e^{\frac{\beta(\omega_{k,j}-T(n))}{T(n)}}\right]^2}\right]\frac{\beta}{T(n)}e^{\frac{\beta(\omega_{k,j}-T(n))}{T(n)}} \tag{28}$$

$$\frac{\partial u}{\partial \omega_{k,j}} = \frac{2\alpha}{T(n)}\left(\frac{\omega_{k,j} - T(n)}{T(n)}\right)e^{-\alpha(\frac{\omega_{k,j}-T(n)}{T(n)})^2} \tag{29}$$

$$\frac{\partial^2 u}{\partial \omega_{k,j}\partial T(n)} = \frac{-2\alpha T(n)^2 - 4\alpha\omega_{k,j} + 4\alpha T(n)}{T(n)^3}e^{-\alpha(\frac{\omega_{k,j}-T(n)}{T(n)})^2} + 4\alpha^2\frac{\omega_{k,j} - T(n)}{T(n)}e^{-\alpha(\frac{\omega_{k,j}-T(n)}{T(n)})^2}\left(\frac{\omega_{k,j} - T(n)}{T(n)}\right)\left(\frac{\omega_{k,j}}{T(n)^2}\right) \tag{30}$$

$\frac{\partial^2 g(i)}{\partial \omega_{k,j}\partial T(n)}$, its expression is too complex, so it will not be expressed in this article.

The convergence condition is $\Delta T < T^2 T_i$. $i$ is decomposition level. The initial threshold of wavelet denoising is $T = q^2\sqrt{2\log(N)}$. $N$ is the number of observation data, and $q^2$ is the noise variance, which can be estimated by the median value of the absolute value at each level.

*3.6. Denoising Process*

The flow chart of adaptive wavelet denosing algorithm is shown in Figure 16.

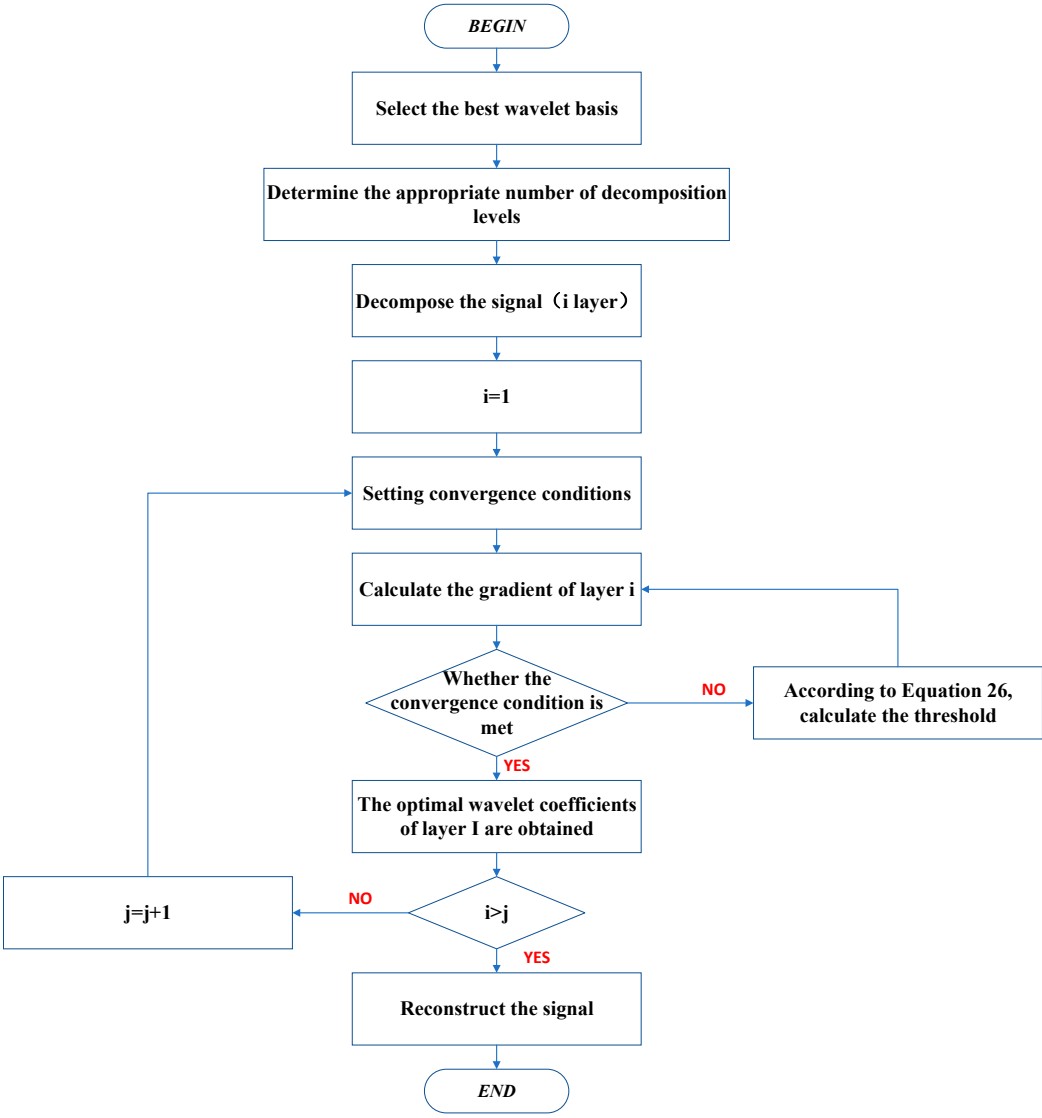

**Figure 16.** New wavelet threshold denoising process.

## 4. Results and Discussion

In order to verify the performance of the proposed de-noising method, computer-generated white Gaussian noise is added to the known magnetic signals and compared with the classical algorithm. From the perspective of reconstructed signal quality, many quantitative parameters can be used to evaluate the performance of denoising process. In this case, the following parameters are compared:

Signal-to-noise ratio (*SNR*):

$$SNR = 10\ln\left[\frac{\sum_n x^2(n)}{\sum_n [x(n) - \hat{x}(n)]^2}\right] \tag{31}$$

where $\hat{x}(n)$ is the denoised signal, and $x(n)$ is the original signal. The constant, *N*, is the number of samples composing the signals.

Root-mean-square error (*RMSE*):

$$RMSE = \sqrt{\frac{\sum\limits_{n}[x(n) - \hat{x}(n)]^2}{length(n)}} \tag{32}$$

where *n* records the length of each element in the decomposed wavelet coefficient matrix.

As can be seen from Figure 17, the denoising results show that the denoising effect of the new method is better than that of the conventional threshold denoising method. Compared with model(xu2018), it can be seen that the denoising effect of this paper is still good. According to Table 3, the SNR value of the adaptive threshold is 96.3, and the RMSE value is 0.2011, indicating that the effect is better than that of the conventional threshold method and model (xu2018). It can be seen from Table 4 that the SNR value of the constructed threshold function is higher than that of the soft threshold function, hard threshold function, and model (xu2018). It shows the superiority of the proposed method. The number of decomposition layers is 5.

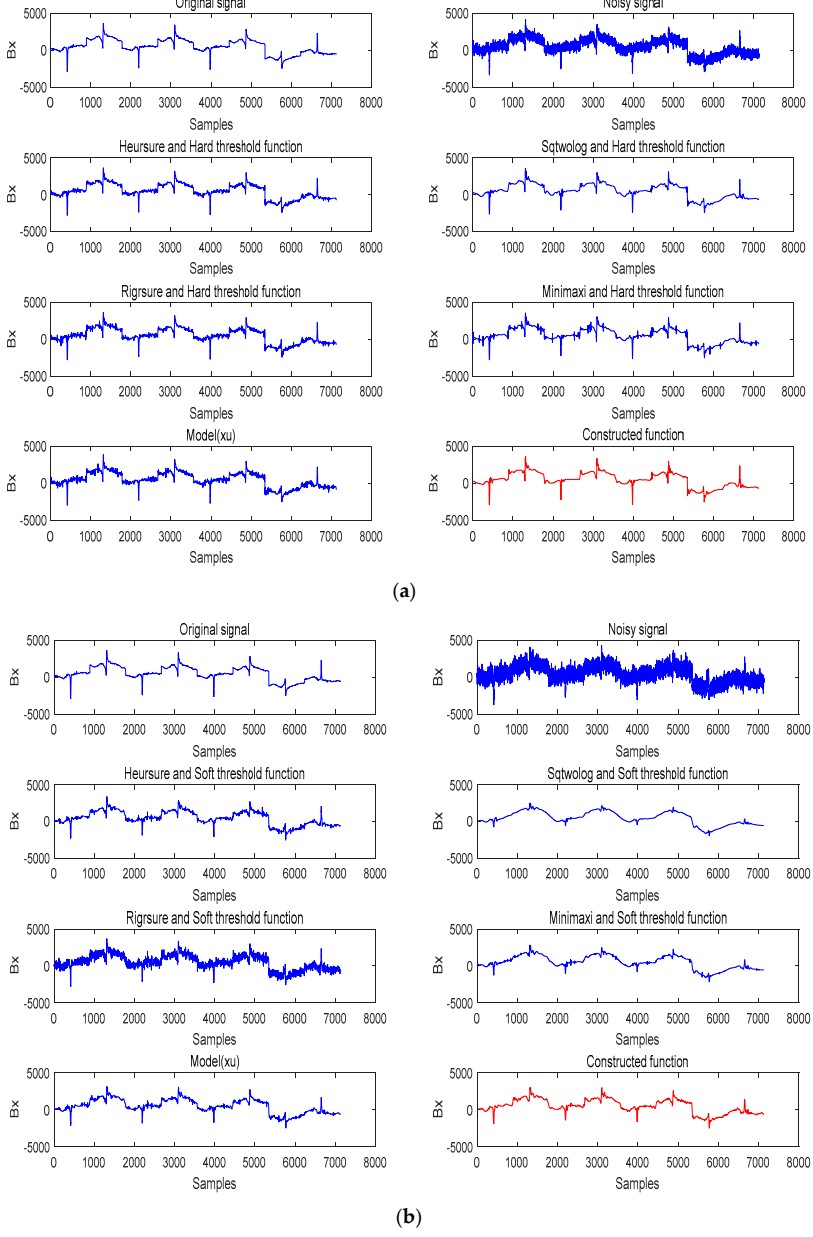

**Figure 17.** Comparison of signal processing results (**a**) hard threshold function; (**b**) soft threshold function.

**Table 3.** Conventional wavelet threshold denoising method.

| Denoising Index | Threshold Function | Wavelet Basis | Wavelet Threshold Selection Method | | | |
|---|---|---|---|---|---|---|
| | | | Sqtwolog | Rigrsure | Heursure | Minimaxi |
| SNR | Hard threshold function | DB7 | 91.3 | 89.88 | 92.62 | 93.98 |
| RMSE | | | 0.2646 | 0.2841 | 0.2477 | 0.2314 |
| SNR | | COIF4 | 89.71 | 91.18 | 92.83 | 88.52 |
| RMSE | | | 0.2866 | 0.2671 | 0.2413 | 0.293 |
| SNR | Soft threshold function | DB7 | 92.4 | 94.18 | 95.73 | 90.71 |
| RMSE | | | 0.2491 | 0.2284 | 0.221 | 0.2769 |
| SNR | | COIF4 | 91.33 | 95.29 | 94.71 | 89.43 |
| RMSE | | | 0.263 | 0.2234 | 0.2251 | 0.2884 |

**Table 4.** Constructed wavelet threshold denoising method.

| Denoising Index | Wavelet Basis | Denoising Method | |
|---|---|---|---|
| | | xu(2018) | Constructed Threshold Function |
| SNR | DB7 | 96.72 | 100.74 |
| RMSE | | 0.207 | 0.1651 |
| SNR | COIF4 | 95.89 | 98.16 |
| RMSE | | 0.2115 | 0.1867 |

## 5. Conclusions

In the process of three-dimensional force and magnetic signal processing, the denoising effect of the signal plays an important role in the feature extraction and analysis of the force and magnetic signal. Firstly, the most appropriate wavelet basis function and decomposition layer number are selected through analysis and calculation. In addition, the new method of a new threshold combined with a new threshold function is used to denoise the analog signal and the measured vibration signal. It shows that the satisfactory noise reduction effect can be achieved by setting the appropriate threshold combined with the new threshold function, which can reflect the real vibration characteristics and retain the high-frequency characteristics of the signal.

According to the simulation results of the analog signal, the new threshold and the improved threshold function can effectively improve the signal-to-noise ratio, reduce the root mean square error, suppress the high-frequency noise, and recover the real useful signal in the signal. According to the measured magnetic signal processing, the improved noise reduction algorithm has a good noise reduction effect in magnetic signal processing.

**Author Contributions:** Conceptualization, X.L. and K.L.; methodology, K.L.; software, J.Z.; validation, X.L. and G.H.; formal analysis, X.L.; investigation, X.L.; resources, X.L.; data curation, G.H.; writing—original draft preparation, X.L.; writing—review and editing, J.Z.; visualization, K.L.; supervision, X.L.; project administration, X.L.; funding acquisition, K.L. All authors have read and agreed to the published version of the manuscript.

**Funding:** This research was funded by the National Natural Science Foundation of China grant number 51674212.

**Data Availability Statement:** The datasets generated and/or analyzed during the present study are available from the corresponding author on reasonable request.

**Conflicts of Interest:** The authors declare no conflict of interest.

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
