# Peer review of "Research on Improved Wavelet Threshold Denoising Method for Non-Contact Force and Magnetic Signals"

_electronics, doi:10.3390/electronics12051244_

Round 1
Reviewer 1 Report
This paper deals with wavelet-transform-based denoising of three-dimensional force and magnetic signals. The authors base their denoising approach on coefficient thresholding using a Bayesian threshold. Although the paper considers potential practical applications, the theory is poorly written and significant revisions are necessary. The technical quality of the paper’s content is low and must be considerably improved. The current version of the paper is not ready even for review. Too many technical shortcomings make the paper currently not readable. The authors do not adequately elaborate on the basic ideas and should significantly improve the presentation. Moreover, they should explain and prove the advantages of their work, particularly in comparison with widely-used denoising techniques.
More specific comments are given next:
- The authors should clarify original contributions and relations with the state-of-the-art in signal denoising.
- The authors must strengthen numerical results and provide a comprehensive comparative analysis of their methodology versus state-of-the-art techniques, to justify the introduction of a new denoising technique.
- Mathematical formulas and equations are erroneous, contain unexplained entities, do not completely clarify fundamental ideas, and sometimes, the equation formatting is not adequate. Examples that illustrate these assessments are numerous, and the Reviewer cannot comprehensively cover all of them in this comment. Some examples include: on page 2, the paragraph above (1), xn, L2(R) represent inadequate mathematical notation. In (1), it is not clear what is summation index and the limits of summations are, and the linear combination in (1) is not clear.
- Figure 1 is too large and variables are not written in math mode.
- Why Section 3, Results and Discussion, contains section 3.3 which presents the theory. This subsection must be moved from Section 3. Why do you present the genetic algorithm in this Section?
- Throughout the paper, the authors do not use math mode for equations, which considerably affects the readability of the paper.
- Equation (26) is not clear.
- The genetic algorithm description is poor and must be improved. The algorithm should be also presented in form of a pseudo-code or flow-chart.
- The authors should significantly improve their English.
Author Response
This paper deals with wavelet-transform-based denoising of three-dimensional force and magnetic signals. The authors base their denoising approach on coefficient thresholding using a Bayesian threshold. Although the paper considers potential practical applications, the theory is poorly written and significant revisions are necessary. The technical quality of the paper’s content is low and must be considerably improved. The current version of the paper is not ready even for review. Too many technical shortcomings make the paper currently not readable. The authors do not adequately elaborate on the basic ideas and should significantly improve the presentation. Moreover, they should explain and prove the advantages of their work, particularly in comparison with widely-used denoising techniques.
More specific comments are given next:
- The authors should clarify original contributions and relations with the state-of-the-art in signal denoising.
Response: Thank you for your good comments. Through the investigation of the latest literature, I find that it is outdated to use Bayesian method to determine the threshold value. Therefore, I adopt the latest adaptive threshold determination method, in other words, Stein's unbiased estimation method to determine the threshold. And a lot of revisions have been made to the article. Compared with the latest denoising methods, the superiority and novelty of this method are illustrated.
- The authors must strengthen numerical results and provide a comprehensive comparative analysis of their methodology versus state-of-the-art techniques, to justify the introduction of a new denoising technique.
Response: Thank you for your useful comments. This paper reproduces the threshold denoising method in reference 17. As shown in Figure 15, the denoising model in reference 17 is compared with the denoising result of the model established in this paper. It is found that the denoising result in this paper is better than the denoising result of model (xu2018), and better than the conventional wavelet threshold denoising method.
- Mathematical formulas and equations are erroneous, contain unexplained entities, do not completely clarify fundamental ideas, and sometimes, the equation formatting is not adequate. Examples that illustrate these assessments are numerous, and the Reviewer cannot comprehensively cover all of them in this comment. Some examples include: on page 2, the paragraph above (1), xn, L2(R) represent inadequate mathematical notation. In (1), it is not clear what is summation index and the limits of summations are, and the linear combination in (1) is not clear.
Response: Thank you for your meaningful comments. Due to some errors in the second section of the paper, in order to express the wavelet threshold denoising process more clearly, I simplified the denoising process.
- Figure 1 is too large and variables are not written in math mode.
Response: Thank you for your farsighted comments. I've made Figure 1 smaller. In addition, Figure 1 is uneditable; it cannot be written in math mode.
- Why Section 3, Results and Discussion, contains section 3.3 which presents the theory. This subsection must be moved from Section 3. Why do you present the genetic algorithm in this Section?
Response: Thank you for your insightful comments. I have deleted the genetic algorithm, adjusted the short circuit according to the structure, and readjusted the title.
- Throughout the paper, the authors do not use math mode for equations, which considerably affects the readability of the paper.
Response: Thank you for your beneficial comments. I don't know what math mode means. In China, we use MathType software to edit formulas. Formulas exist in the form of pictures in word.
- Equation (26) is not clear.
Response: Thank you for your significant comments. I have removed the Bayesian thresholding method and genetic algorithm and adopted the latest adaptive thresholding method.
- The genetic algorithm description is poor and must be improved. The algorithm should be also presented in form of a pseudo-code or flow-chart.
Response: Thank you for your helpful comments. Since the Bayesian thresholding method is outdated, I adopt the latest adaptive thresholding method and remove the genetic algorithm.
- The authors should significantly improve their English.
Response: Thank you for your provident comments. I have checked the English of the paper in detail and improved it.
Reviewer 2 Report
This paper presents a denoising technique based on an improved wavelet threshold, which is applied to the non-contact force and magnetic signals. The traditional threshold methods have shortcomings that have been addressed by the authors in this manuscript. The authors showed their method performs better with the help of “SNR” and “RMSE”. As far as I am concerned, the results are good and interesting. However, the presentation can be improved.
1. There are many places where the carriage return \r should be required, for instance in line 54 “linear combination of basis functions \r …”, line 130 \r required after Fig., in the caption of Figure 15, etc.
2. All the symbols used in the equations and the text are not matching. E.g. The symbols in equation (2) and their explanation has different style of symbols such as , etc. This kind of mismatching is almost in every equation. There are also some of the symbols explained in the test which is not a part of the equations such as in line 62, Z is not referring to z and in line 68, “I”.
3. Two abbreviations "SWWT" and "iswt" appeared in the Mathematical background. The full names should be provided. Otherwise, the readers may have no idea of the meaning.
4. In Table 2, the full name of TSC should be provided. In the keyboard specification, it should be mentioned 14 “keys”.
5. Almost all the figures are of very bad quality, they must be of vector graphics so that the reader can visualize them.
6. Few statements seem to be incomplete or require rephrasing such as in line 145 “Select db7….. verify”, line 158 “On the other hand …….. avoided”, line 275 “The objection ……” etc.
7. There are some spelling mistakes such as in line 178 change is written as chang, in equation (8), length is written as lenth(l), etc. The manuscript must be read carefully.
8. In equation (10), is not defined. How and from where does it come from?
9. Some of the figures were not referred to and, in the text, it is mentioned like “figure above” or figure “below”. These statements need to be replaced with proper referring to the corresponding figures, such as in lines 126, 197, and 230.
10. The caption of the tables and figures needs to be more elaborative.
11. The details of the different parameters selected for this application are not mentioned in the paper. There must be a table, which shows the value of all the parameters
12. Figure 10 seems similar output of denoising for different decomposition levels. Please explain why this is so.
13. There are other data-driven adaptive thresholding techniques used in many computer vision problems such as work done in “Joint Geometry and Color Point Cloud Denoising Based on Graph Wavelets”. There should be a comparison, on why this wavelet performs better from such thresholding techniques.
Author Response
This paper presents a denoising technique based on an improved wavelet threshold, which is applied to the non-contact force and magnetic signals. The traditional threshold methods have shortcomings that have been addressed by the authors in this manuscript. The authors showed their method performs better with the help of “SNR” and “RMSE”. As far as I am concerned, the results are good and interesting. However, the presentation can be improved.
- There are many places where the carriage return \r should be required, for instance in line 54 “linear combination of basis functions \r …”, line 130 \r required after Fig., in the caption of Figure 15, etc.
Response: Thank you for your good comments. I have adjusted the structure of the paragraphs and added carriage returns where appropriate
- All the symbols used in the equations and the text are not matching. E.g. The symbols in equation (2) and their explanation has different style of symbols such as , etc. This kind of mismatching is almost in every equation. There are also some of the symbols explained in the test which is not a part of the equations such as in line 62, Z is not referring to z and in line 68, “I”.
Response: Thank you for your useful comments. I have unified the letters of the text with the letters of the formula. The letters in the text were then corrected.
- Two abbreviations "SWWT" and "iswt" appeared in the Mathematical background. The full names should be provided. Otherwise, the readers may have no idea of the meaning.
Response: Thank you for your meaningful comments. I have explained “iswt” and deleted “SWWT”.
- In Table 2, the full name of TSC should be provided. In the keyboard specification, it should be mentioned 14 “keys”.
Response: Thank you for your farsighted comments. I have explained "TSC" in the article. In the keyboard specification, I've changed 14 to the 14 keys.
- Almost all the figures are of very bad quality, they must be of vector graphics so that the reader can visualize them.
Response: Thank you for your insightful comments. I have replaced the pictures in the text with pictures with a resolution of less than 300dpi with clearer ones. 300dpi), so that readers can see them more clearly.
- Few statements seem to be incomplete or require rephrasing such as in line 145 “Select db7….. verify”, line 158 “On the other hand …….. avoided”, line 275 “The objection ……” etc.
Response: Thank you for your beneficial comments. I have corrected these sentences in the paper.
- There are some spelling mistakes such as in line 178 change is written as chang, in equation (8), length is written as lenth(l), etc. The manuscript must be read carefully.
Response: Thank you for your significant comments. I have checked the paper carefully and corrected some misspelled words
- In equation (10), is not defined. How and from where does it come from?
Response: Thank you for your helpful comments. Because the threshold selection method in the paper is outdated, I used the latest adaptive threshold selection method, so I deleted Equation 10.
- Some of the figures were not referred to and, in the text, it is mentioned like “figure above” or figure “below”. These statements need to be replaced with proper referring to the corresponding figures, such as in lines 126, 197, and 230.
Response: Thank you for your provident comments. I have reworked these sentences.
- The caption of the tables and figures needs to be more elaborative.
Response: Thank you for your excellent comments. I have explained the important figures and tables in the paper in more detail so that readers can understand them better.
- The details of the different parameters selected for this application are not mentioned in the paper. There must be a table, which shows the value of all the parameters.
Response: Thank you for your standout comments. I have explained the denoising parameters of wavelet threshold denoising in detail, as shown in Table 3. In addition, the denoising parameters used in FIG. 4 are explained.
- Figure 10 seems similar output of denoising for different decomposition levels. Please explain why this is so.
Response: Thank you for your distinguished comments. As can be seen from Figure 8 and Figure 9, when the number of decomposition layers is 5, 6, 7, 8 and 9, their SNR value (196.7) and RMSE value(0.256) are relatively close. it indicates that their denoising effect is relatively close, so figure 10 seems similar output of denoising for different decomposition levels.
- There are other data-driven adaptive thresholding techniques used in many computer vision problems such as work done in “Joint Geometry and Color Point Cloud Denoising Based on Graph Wavelets”. There should be a comparison, on why this wavelet performs better from such thresholding techniques.
Response: Thank you for your outstanding comments. Through the investigation of the latest literature, I found that the Bayesian threshold selection method is outdated, so I adopted the latest adaptive threshold method. By reproducing Xu's model, the model results are compared with the denoising effect of this paper, and it is found that the denoising effect of this method is better than Xu's method.